# High-throughput synapse-resolving two-photon fluorescence microendoscopy for deep-brain volumetric imaging in vivo

Guanghan Meng[1,2,3,4,5], Yajie Liang[2], Sarah Sarsfield[6], Wan-chen Jiang[2], Rongwen Lu[2], Joshua Tate Dudman[2], Yeka Aponte[3,6], Na Ji[1,4,5,7]*

[1]Department of Molecular and Cell Biology, University of California, Berkeley, United States; [2]Janelia Research Campus, Howard Hughes Medical Institute, Ashburn, United States; [3]Solomon H Snyder Department of Neuroscience, Johns Hopkins University School of Medicine, Baltimore, United States; [4]Department of Physics, University of California, Berkeley, United States; [5]Helen Wills Neuroscience Institute, University of California, Berkeley, United States; [6]Intramural Research Program, Neuronal Circuits and Behavior Unit, National Institute on Drug Abuse, National Institutes of Health, Baltimore, United States; [7]Molecular Biophysics and Integrated Bioimaging Division, Lawrence Berkeley National Laboratory, Berkeley, United States

**Abstract** Optical imaging has become a powerful tool for studying brains in vivo. The opacity of adult brains makes microendoscopy, with an optical probe such as a gradient index (GRIN) lens embedded into brain tissue to provide optical relay, the method of choice for imaging neurons and neural activity in deeply buried brain structures. Incorporating a Bessel focus scanning module into two-photon fluorescence microendoscopy, we extended the excitation focus axially and improved its lateral resolution. Scanning the Bessel focus in 2D, we imaged volumes of neurons at high-throughput while resolving fine structures such as synaptic terminals. We applied this approach to the volumetric anatomical imaging of dendritic spines and axonal boutons in the mouse hippocampus, and functional imaging of GABAergic neurons in the mouse lateral hypothalamus in vivo.
DOI: https://doi.org/10.7554/eLife.40805.001

*For correspondence:
jina@berkeley.edu

## Introduction

Neural circuits and neurons therein are three-dimensional (3D) structures that may extend over hundreds or thousands of microns. Understanding their operations requires monitoring their activity at synaptic and cellular resolution in 3D at image rates that capture all activity events. In brains that strongly scatter light (*i.e.*, all known adult mammalian brains), the image depth of conventional multiphoton microscopy is limited to 1–2 millimeters (*Ji et al., 2016*). To reach depths beyond a few millimeters, the only practical methods currently available are based on microendoscopy, in which miniature probes, such as gradient refractive index (GRIN) lenses, are embedded within the brain to relay microscopic images of the neurons below (*Jung and Schnitzer, 2003*; *Jung et al., 2004*; *Levene et al., 2004*). Such systems have been used for fluorescence activity imaging of neurons expressing genetically encoded calcium indicators in a variety of deep nuclei in the mouse brain (*Ghosh et al., 2011*; *Li et al., 2017*; *Ziv et al., 2013*; *Bocarsly et al., 2015*; *Jennings et al., 2015*; *Kitamura et al., 2015*; *Pinto and Dan, 2015*; *Harrison et al., 2016*; *Okuyama et al., 2016*; *Kamigaki and Dan, 2017*; *Kitamura et al., 2017*; *McHenry et al., 2017*; *Roy et al., 2017*);

however, the spatial resolution has been limited to the cellular level and synaptic structures such as dendritic spines or axonal boutons were not resolved in these studies.

In addition to its limited spatial resolution, compared to imaging superficial brain regions with a conventional microscope, experiments involving microendoscopy imaging of deep structures face additional challenges. First, surgeries to remove brain tissue and embed endoscope probes typically have lower success rates compared to conventional craniotomies in which no brain tissue is removed. Second, many microendoscopy experiments involve behaviors or brain states with large variability that either cannot be evoked with high fidelity over repeated trials or exhibit short-term adaptation. Finally, research interests are often centered on specific cell types (*e.g.*, inhibitory subtypes [*Pinto and Dan, 2015*] or cholinergic neurons [*Harrison et al., 2016*]) that are usually sparsely distributed in 3D. All these factors limit the yield of experiments using conventional two-photon microendoscopes in which neurons are imaged one cellular layer at a time. For these reasons, a volumetric microendoscopic imaging method, which can simultaneously probe the activity of neural ensembles in 3D while maintaining synapse-resolving spatial resolution, would be greatly beneficial for interrogating neural circuit functions at depth but has yet to be demonstrated for in vivo volumetric imaging of deep brain structures.

In contrast to the paucity of high-throughput, high-resolution volumetric imaging solutions for microendoscopy, a suite of methods has been developed to image superficial structures at high speed with conventional microscopy (*Ji et al., 2016*). Recently, we developed a Bessel focus scanning method that, when combined with two-photon fluorescence microscopy (*Denk et al., 1990*), enabled fast volumetric imaging with synaptic resolution (*Lu et al., 2017*) in vivo. By scanning an axially extended focus in two dimensions (2D), we obtained projection images of 3D volumes that increased imaging throughput and reduced data size by 10-to-100-fold. The elongation of the focus in the axial direction also makes imaging resistant to axial motion artifacts, making it ideally suited to the study of neural circuit functions in awake, behaving animals. Here, we showed that high-throughput, high-resolution volumetric imaging of neurons and neural activity in deeply buried nuclei is achievable by combining Bessel focus scanning with microendoscopy. We systematically evaluated and identified commercially available GRIN lenses through which high-quality Bessel foci can be generated. We demonstrated in brain slices ex vivo and in the mouse brain in vivo that volumetric imaging can be achieved with microendoscopy while maintaining synapse-resolving resolution for anatomical and functional imaging of neurons at depth.

## Results

### A two-photon excitation fluorescence microendoscopy system with Bessel focus scanning

The schematics of a two-photon excitation fluorescence microendoscopy system with a Bessel focus scanning module is shown in *Figure 1A* (see Materials and methods). Based on a standard two-photon excitation fluorescence microscope equipped with a pair of galvo scanners, the endoscopy part of the microscope was realized by having a GRIN lens relay the focus of a conventional microscope objective from the image side to the sample side (*Figure 1B*). Due to the small size of the mouse brain, we only considered GRIN lenses with 0.5 mm or 1 mm diameters and a numerical aperture (NA) of 0.5. For each diameter, the GRIN lens is either a singlet with NA = 0.5 on both sides or a doublet with an object NA of 0.5 and an image NA of 0.2 or 0.1 (*Figure 1C*). With part of the GRIN lens assembly having a lower NA, doublet GRIN lenses have lower intrinsic aberrations than singlets of similar lengths. A 0.45 NA microscope objective was used in conjunction with 0.5-NA singlet GRIN lenses, whereas a 0.2 NA objective was used with doublet GRIN lenses to match their NA on the image side.

The Bessel focus scanning module (blue rectangle, *Figure 1A*) was described in detail previously (*Lu et al., 2017*). Briefly, a liquid crystal spatial light modulator (SLM) was used to impart specific phase patterns onto the excitation light. As a result, after passing through lenses and being spatially filtered by an annular mask, the excitation light formed a ring-like pattern at the back focal plane of the microscope objective, which in turn generated an axially-elongated, Bessel-like focus in the focal region. A pair of flip mirrors allowed an easy switch from the Bessel imaging mode back to the conventional illumination scheme with a Gaussian beam profile at the back focal plane and a

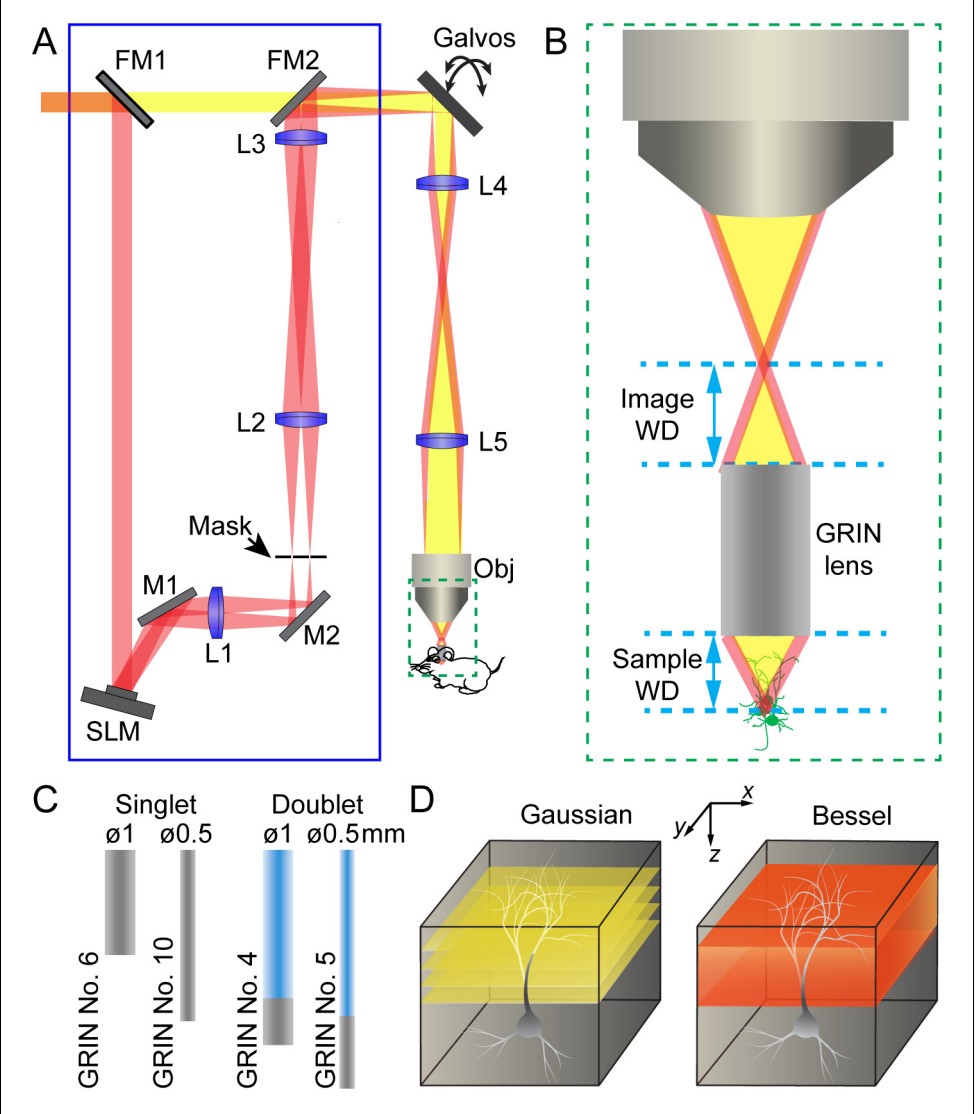

**Figure 1.** Schematics of two-photon excitation fluorescence microendoscopy with a Bessel focus scanning module. (**A**) A Bessel focus scanning module (blue rectangle) made of a spatial light modulator (SLM), a lens (L1), and an annular mask is incorporated into a two-photon fluorescence microendoscopy system. L2-L5: lenses for optical conjugation; M1, M2: turning mirrors; Galvos: x and y galvanometers; Obj: microscope objective; FM1, FM2: flip mirrors to switch between Gaussian (yellow) and Bessel (red) beam paths. (**B**) Enlarged view of a GRIN lens relaying the focus of the objective into a focus inside the brain. Image WD: image working distance, distance between the objective focus and the top surface of GRIN lens; Sample WD: sample working distance, distance between the bottom of the GRIN lens and the sample structure in focus. (**C**) Two singlet and two doublet GRIN lenses selected for further characterization. (**D**) To image a volume requires multiple 2D scans of a Gaussian focus but a single 2D scan with a Bessel focus.

DOI: https://doi.org/10.7554/eLife.40805.002

more axially restricted focus (referred to as the Gaussian mode hereafter). In the Gaussian mode,, multiple planes were scanned to acquire anatomical or functional information from a brain volume; whereas in the Bessel mode, volumetric information was obtained via a single 2D scan (*Figure 1D*). The resulting fluorescence was collected by a photomultiplier tube following the detection scheme of standard two-photon excitation microscopy.

## Characterization and identification of GRIN lenses with optimal imaging performance

GRIN lenses have intrinsic aberrations (*Lee and Yun, 2011*; *Bortoletto et al., 2011*), which limit the resolution and field of view (FOV) of microendoscopy under the conventional Gaussian mode (*Wang and Ji, 2013*; *Wang and Ji, 2012*). To generate Gaussian and Bessel foci of the highest quality, GRIN lenses with minimal aberrations are required. However, a systematic evaluation of the imaging performance of commercially available GRIN lenses had not been undertaken. We therefore characterized GRIN lenses commonly used for in vivo brain imaging to identify GRIN lenses with optimal imaging performance for both Gaussian and Bessel modes.

To evaluate the imaging qualities of GRIN lenses, 2-μm-diameter fluorescent beads were fixed on a coverslip and immersed in water below each GRIN lens. With the distance between the microscope objective and the GRIN lens fixed, we moved the bead sample axially until the beads were in focus. The distance between the microscope objective focus and the GRIN lens top surface was defined as image working distance (WD), while the distance between the bottom surface of the GRIN lens and the bead sample was measured and defined as the sample WD (*Figure 1B*). Such measurements informed on the axial focal shifts inside the brain while the brain and the implanted GRIN lens were moved together axially during in vivo imaging experiments. At each sample/image WD, we measured the axial full width at half maximum (FWHM) of the 2 μm beads at the center of the FOV, and this measurement served as an indicator of spatial resolution. The optimal GRIN lens should have small axial FWHM, or equivalently, high resolution over a large FOV (with the edges of the FOV defined as where fluorescence signal from 2-μm-diameter beads dropped to 10% of the signal at the FOV center).

Of the ten GRIN lenses tested (*Figure 2*), three were found to have the best axial FWHM – FOV combinations (green rectangles, *Figure 2*): a 1-mm-diameter, 8.1-mm-long doublet GRIN lens with 370-μm-diameter FOV ('1 mm doublet', GRIN No. 4); a 0.5-mm-diameter, 9.9-mm-long doublet GRIN lens with 220-μm-diameter FOV ('0.5 mm doublet', GRIN No. 5); and a 1-mm-diameter, 4.4-mm-long singlet GRIN lens with 510-μm-diameter FOV ('1 mm singlet', GRIN No. 6). For all three GRIN lenses, the axial FWHMs were below 15 μm over most of the sample/image WDs, substantially less than the other GRIN lenses tested (but still larger than the diffraction-limited value at 0.5 NA, 6.3 μm, indicating the existence of intrinsic aberrations). In addition to these best performing lenses, we chose an additional GRIN lens for further characterization, a 0.5-mm-diameter, 7.1-mm-long singlet GRIN lens with 310-μm-diameter FOV and an axial FWHM between 18 μm and 32 μm ('0.5 mm singlet', red rectangle, *Figure 2*, GRIN No. 10). Even though it had larger aberrations, this GRIN lens had the lowest cost of all GRIN lenses tested and the largest FOV for 0.5-mm-diameter GRIN lenses. These four GRIN lenses (*Figure 1C*) were selected for further experiments.

Applying different binary phase patterns to the SLM in the Bessel module (*Lu et al., 2017*), we generated axially extended foci of varying NAs and discovered that Bessel foci of 0.3 NA had the best performance, yielding higher quality images of fluorescent beads both in the center and towards the edges of the FOV (*Figure 3*). By changing the incident beam size onto the SLM, the axial FWHM of the Bessel focus could be easily adjusted from 40 μm to 120 μm. Furthermore, we found 0.3-NA Bessel foci to have higher lateral resolution than 0.5-NA Gaussian foci (*Welford, 1960*; *Sheppard and Wilson, 1979*) at the center of the FOV, as indicated by the lateral point spread functions measured from 0.2-μm-diameter fluorescent beads (*Figure 4*) as well as superior performance when imaging 2-μm-diameter fluorescent beads away from the FOV center (*Figure 5*).

For all four GRIN lenses selected for further characterization, Bessel foci consistently resolved 2-μm-diameter beads at the center of their FOV (*Figure 5*, *Figure 6*). Their FOV sizes, when compared to those of Gaussian foci, varied among different GRIN lenses. For the three GRIN lenses of good quality, the image FOVs obtained with Bessel foci were larger than (1 mm doublet, *Figure 6A*; 1 mm singlet, *Figure 5A*) or equivalent to (0.5 mm doublet, *Figure 5B*) those of Gaussian foci. For example, the 1 mm doublet had a 390-μm-diameter FOV for a Bessel focus of 53 μm axial FWHM, which was larger than the 370 μm FOV of the Gaussian focus (*Figure 6A,B,C*). In contrast, for the 0.5 mm singlet with poorer imaging performance, a Bessel focus of similar axial length had its FOV slightly reduced when compared to that of the Gaussian focus (*Figure 6D,E,F*).

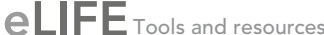

**A** Factory information and measured field of view (FOV) size of GRIN lenses

| | GRIN No. 1 | GRIN No. 2 | GRIN No. 3 | GRIN No. 4 | GRIN No. 5 | GRIN No. 6 | GRIN No. 7 | GRIN No. 8 | GRIN No. 9 | GRIN No.10 |
|---|---|---|---|---|---|---|---|---|---|---|
| **Company** | GRINTECH | GRINTECH | GRINTECH | GRINTECH | GRINTECH | GRINTECH | Inscopix | Inscopix | Inscopix | Go!Foton |
| **Singlet(S)/ Doublet(D)** | D | D | D | D | D | S | S | S | S | S |
| **Sample NA/ Image NA** | 0.5 / 0.1 | 0.5 / 0.1 | 0.5 / 0.2 | 0.5 / 0.2 | 0.5 / 0.2 | 0.5 / 0.5 | 0.5 / 0.5 | 0.5 / 0.5 | 0.5 / 0.5 | 0.5 / 0.5 |
| **Diameter (mm)** | 1.0 | 1.0 | 1.0 | 1.0 | 0.5 | 1.0 | 0.5 | 0.5 | 1.0 | 0.5 |
| **Product Information/ Part number** | -- | -- | NEM-100-25-10-860-DL | NEM-100-25-10-860-DS | NEM-050-25-10-860-DM | NEM-100-25-10-860-S | -- | -- | -- | ILW imaging lens, 1.46 pitch, 550nm, no coating |
| **Length (mm)** | 35 | 36 | 19.93 | 8.1 | 9.89 | 4.36 | 8.4 | 6.1 | 4.0 | 7.1 |
| **FOV diameter (µm)\*** | 250 | 240 | 280 | 370 | 220 | 510 | 270 | 280 | 480 | 310 |

\*Edge of FOV defined as where fluorescence signal from 2-µm-diameter beads drops to 10% of the signal at the center

**B** GRIN No.1   GRIN No.2   GRIN No.3   GRIN No.4   GRIN No.5   GRIN No.6   GRIN No.7   GRIN No.8   GRIN No.9   GRIN No.10

**C** GRIN No.1   GRIN No.2   GRIN No.3   GRIN No.4   GRIN No.5   GRIN No.6   GRIN No.7   GRIN No.8   GRIN No.9   GRIN No.10

Axial FWHM (µm)

Sample WD (µm)

k = -0.09   k = -0.08   k = -0.18   k = -0.17   k = -0.23   k = -1.49   k = -0.94   k = -1.01   k = -1.59   k = -1.45

Image WD (µm)

**Figure 2.** Characterization of ten GRIN lenses. (**A**) Factory information and field of view (FOV) sizes of ten GRIN lenses. (**B**) Maximal intensity projections of 3D image stacks of 2-µm-diameter fluorescent beads obtained under conventional Gaussian focus scanning mode (3D stack was required due to field curvature). Scale bar: 100 µm. (**C**) Top panel: the axial full width at half maximum (FWHM) of 2-µm-diameter beads at different image working distance (WD). Bottom panel: measured (black dot) relationship between image WD and sample WD, with k being the slope of a linear fit (blue line).
DOI: https://doi.org/10.7554/eLife.40805.003

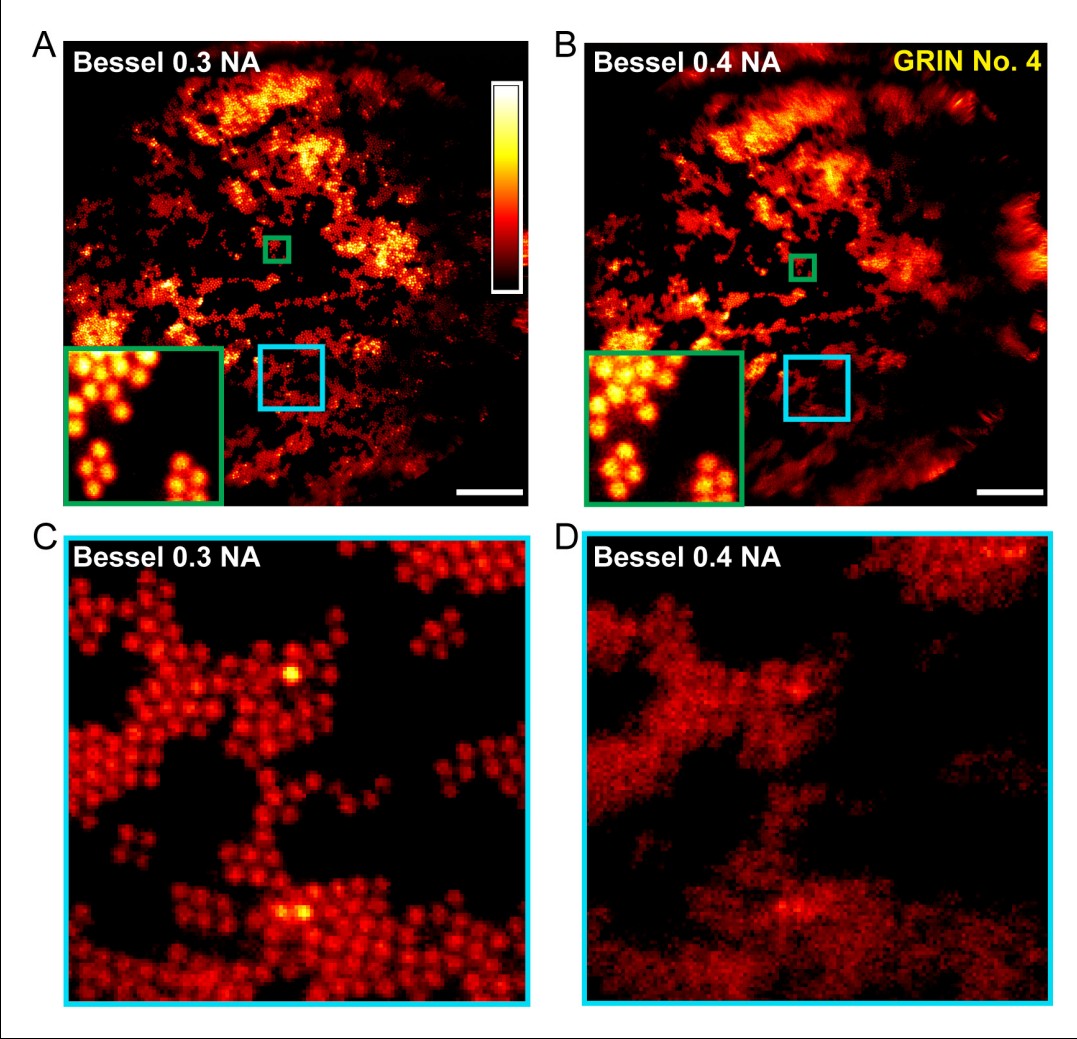

**Figure 3.** 0.3-NA Bessel foci achieves better imaging performance than 0.4-NA Bessel foci. (**A,B**) Images of 2-µm-diameter fluorescent beads obtained with Bessel foci of 0.3 NA, 51 µm axial FWHM and 0.4 NA, 45 µm axial FWHM, respectively. Insets show enlarged views of beads at FOV center (green squares). (**C,D**) Enlarged views of beads away from the FOV center (cyan squares in A and B). Scale bar: 20 µm.
DOI: https://doi.org/10.7554/eLife.40805.004

Furthermore, within the FOV for both Gaussian and Bessel foci, the GRIN lenses of higher quality (**Figure 5**, **Figure 6A**) provided more uniform excitation efficiency than the GRIN lens with lower imaging quality (0.5 mm singlet, **Figure 6D**), as indicated by the variation of 2-µm-diameter bead signal across the FOV (*cf.*, **Figure 6C** and **Figure 6F**). This rapid decrease of bead signal from the center in the 0.5 mm singlet GRIN lens (**Figure 6F**) was caused by its larger off-axis optical aberration, as demonstrated by the signal and the 3D image profiles of 2 µm beads located at the center (cyan squares, **Figure 6A,D**) or closer to the edge of the FOV (~85 µm from center, magenta squares, **Figure 6A,D**) taken with Gaussian foci. At the center of the FOV (upper panels, **Figure 6G, H**), the aberration modes of both 1 mm doublet and 0.5 mm singlet were dominated by spherical aberrations. The circular symmetry of spherical aberrations made them minimally degrade the imaging quality of the Bessel focus formed by annular illumination. As a result, we found the measured lateral FWHMs of the Bessel foci (0.93–1.27 µm, **Figure 4**) to be similar to their diffraction-limited value (1.06 µm) (**Richards and Wolf, 1959**), whereas the lateral resolution of the Gaussian foci (measured FWHMs: 1.02–1.47 µm, **Figure 4**, diffraction-limited value: 0.78 µm) were substantially degraded by aberrations. Away from the FOV center, the 3D bead profile obtained through the 1

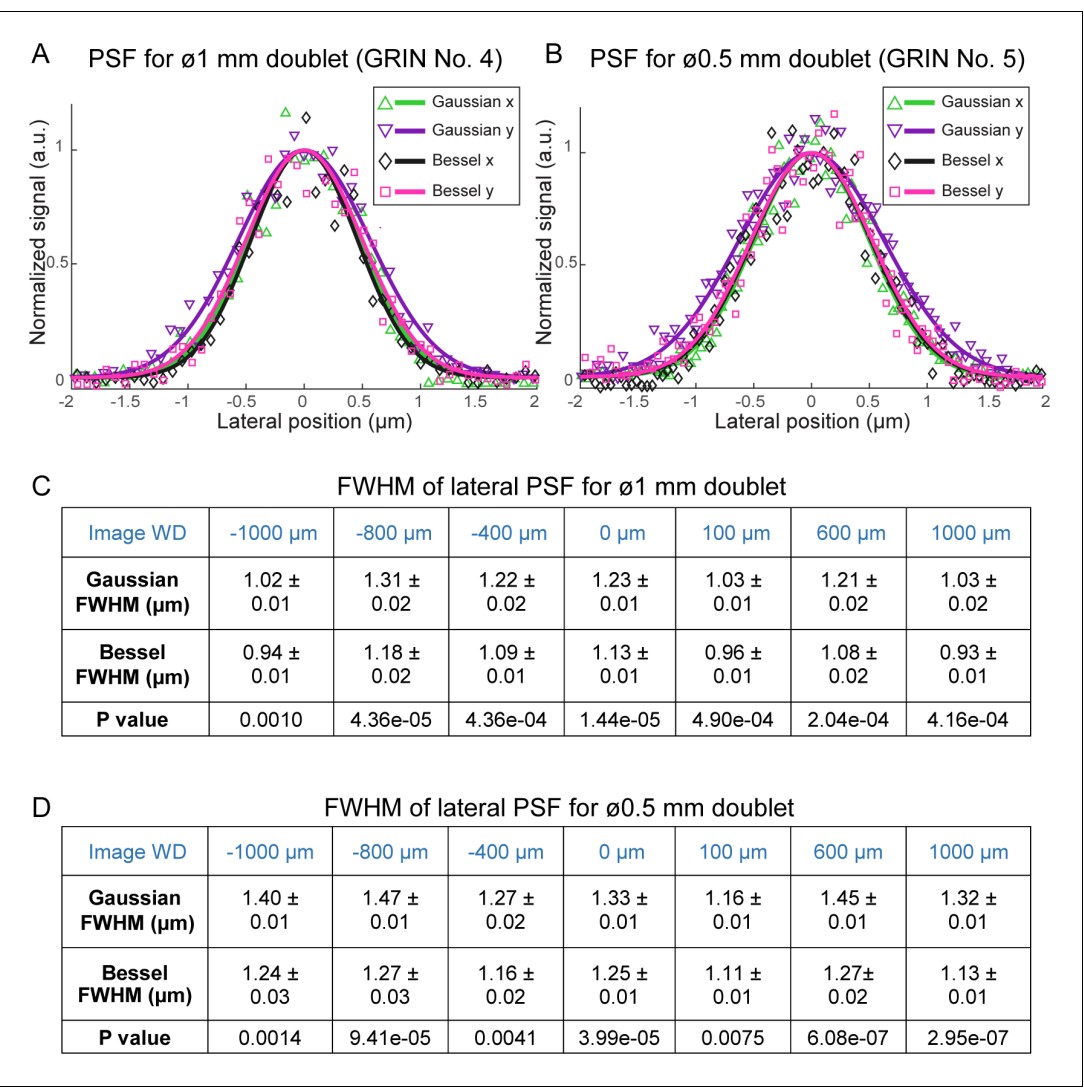

**Figure 4.** 0.3-NA Bessel foci have higher lateral resolution than 0.5-NA Gaussian foci through GRIN lenses. (A,B) Lateral point spread functions (PSFs) of 0.3-NA Bessel foci (axial FWHM: 57 μm) and 0.5-NA Gaussian foci measured with 0.2-μm-diameter fluorescent beads through 1 mm and 0.5 mm doublets, respectively, at 0 μm image WD. PSFs along x and y directions were fitted with a normalized Gaussian function. (C,D) Bessel foci have narrower lateral PSFs than Gaussian foci across the full range of image WDs tested (−1000 μm to 1000 μm) for 1 mm and 0.5 mm doublets. FWHMs: mean ±standard errors of 6–11 measurements. Sample size depends on the number of isolated beads in the acquired images. P values: non-paired t-tests.
DOI: https://doi.org/10.7554/eLife.40805.005

mm doublet maintained similar characteristics to the profile at FOV center, suggesting a similar domination of spherical aberrations (lower panels, *Figure 6G*). In contrast, the off-axis aberrations of the 0.5 mm singlet GRIN lens bore strong characteristics of astigmatism (lower panels, *Figure 6H*), which deteriorated the quality of the Bessel focus more than the Gaussian focus. Therefore, the relative FOV change of Bessel versus Gaussian modes was correlated with the optical aberrations present in the GRIN lenses, with Bessel imaging mode performing better in GRIN lenses with higher imaging qualities.

By carefully characterizing the imaging performance of the four selected GRIN lenses, we found that Bessel focus scanning improves the lateral resolution at the FOV center while providing a comparable, if not larger, FOV compared to conventional Gaussian imaging. These four GRIN lenses have different diameters, lengths, resolutions, FOVs, are suitable for experiments with different goals, and can give us access to different structures throughout the mouse brain. For both Bessel

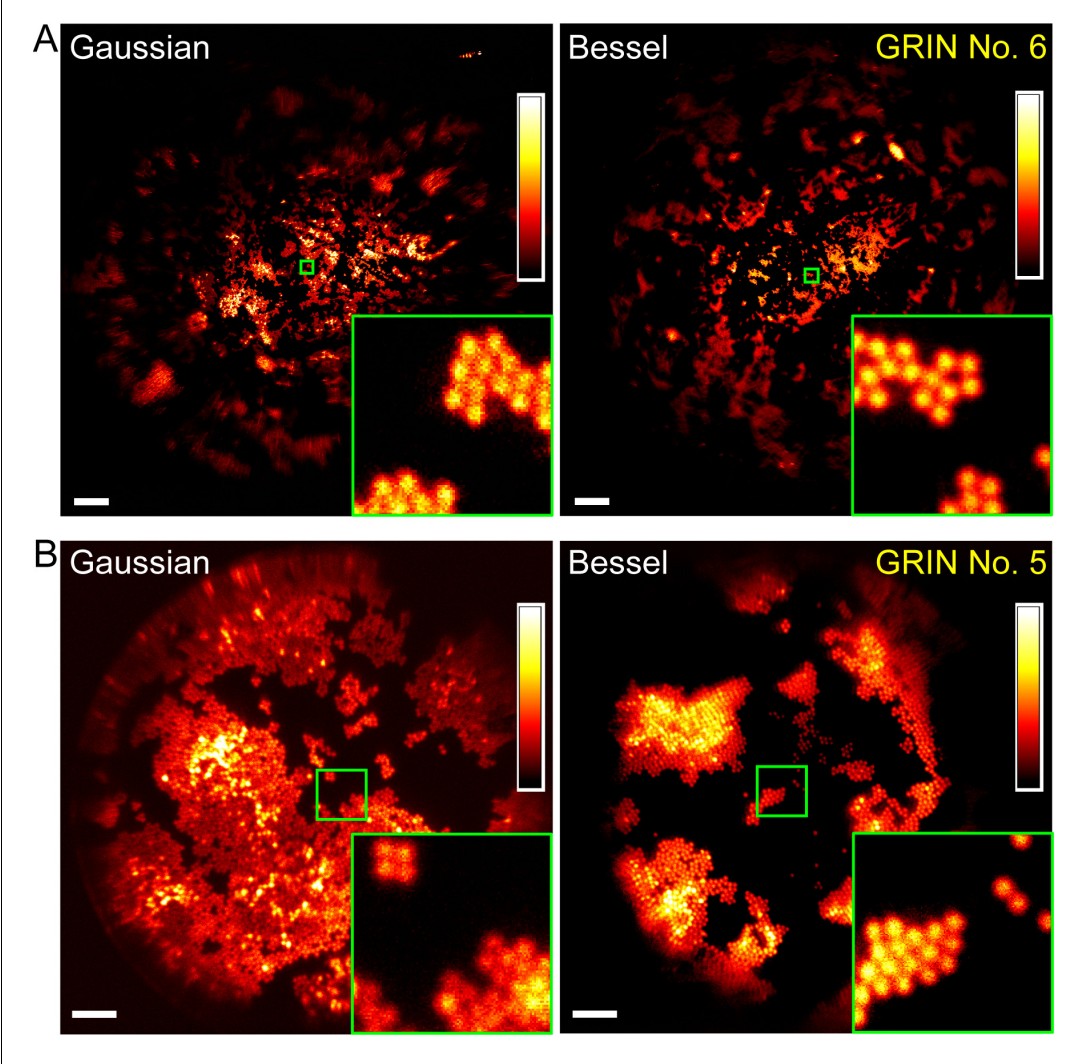

**Figure 5.** Bessel focus scanning produces superior imaging performance to Gaussian focus scanning. (**A**) Through a 1-mm-diameter singlet GRIN lens, Bessel focus scanning has a larger FOV and resolves 2-μm-diameter beads at the central FOV. (**B**) Through a 0.5-mm-diameter doublet GRIN lens, Bessel focus scanning has similar FOV to Gaussian focus scanning and resolves 2-μm-diameter beads at the central FOV. Insets: zoomed-in views of beads at FOV centers (green squares) indicate that Bessel foci have higher lateral resolution than Gaussian foci. Scale bar: 60 μm in (**A**); 20 μm in (**B**).
DOI: https://doi.org/10.7554/eLife.40805.006

and Gaussian modes, if budget and space allow, higher quality GRIN lenses should always be utilized to achieve optimal imaging performance.

## Bessel focus scanning improves microendoscopic imaging throughput with synaptic lateral resolution in brain tissues

After characterizing these GRIN lenses with fluorescent beads, we tested their imaging performance on fixed brain tissue. Two types of samples were imaged: coronal cortical sections from Gad2-ires-Cre mice transduced with AAV-Syn-Flex-GCaMP6s, providing expression of GCaMP6s in GABAergic neurons, and coronal cortical sections from Thy1-GFP-M mice, which provide sparse labeling of pyramidal neurons with GFP. Scanning 0.3-NA Bessel foci with axial FWHM ranging from 53 to 68 μm, we improved microendoscopic imaging throughput substantially while maintaining lateral resolution and FOV through all three best-performing GRIN lenses (*Figure 7*). For the 1 mm singlet (*Figure 7C,D*), a single 2D scan using a 0.3-NA, 68-μm-axial-FWHM Bessel focus captured all 233 interneurons distributed across a 510 μm × 510 μm × 70 μm volume, which required 15 scans of the

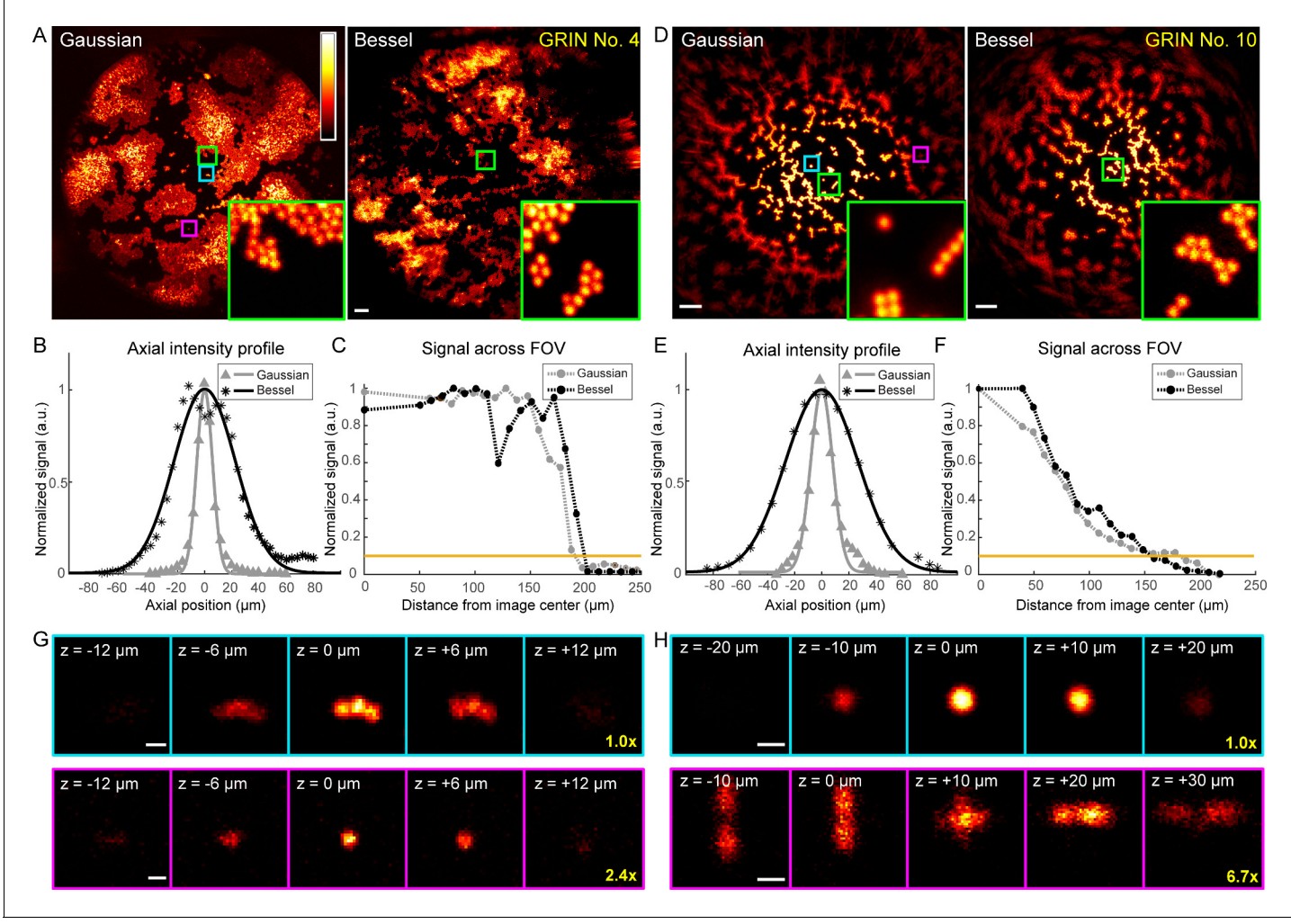

**Figure 6.** Higher quality GRIN lenses produce superior images with both Gaussian and Bessel foci. (**A**) Images of 2-µm-diameter fluorescent beads obtained with a 0.5-NA Gaussian and a 0.3-NA Bessel focus, respectively, through a 1-mm-diameter doublet GRIN lens. Insets: zoomed-in views of beads at the FOV center (green square). (**B**) Axial intensity profiles of the Gaussian and Bessel foci used in (**A**). (**C**) Signal variation across FOVs in (**A**); horizontal orange line indicates 10% of signal at the center, thus the border of the FOV. (**D–F**) Same measurements as (**A–C**), but for a 0.5-mm-diameter singlet GRIN lens of inferior imaging quality. (**G, H**) Gaussian image stacks of beads (upper panel) at FOV center (cyan square in A and D) and (lower panel) away from FOV center (purple square in A and D, 82 µm and 89 µm from center), respectively. Scale bars: 20 µm in (**A, D**), 2 µm in (**G, H**).
DOI: https://doi.org/10.7554/eLife.40805.007

Gaussian focus to cover (*Figure 7C*). Similarly and without compromise in lateral resolution, Bessel focus scanning through this same GRIN lens imaged a 510 µm × 510 µm × 72 µm volume of somata and neurites of sparsely labeled pyramidal neurons that otherwise required 19 Gaussian 2D planes to image (*Figure 7D*). For both 1.0 mm and 0.5 mm doublets (*Figure 7E–H*), the significantly higher lateral resolution of the 0.3-NA Bessel foci allowed dendritic spines of the pyramidal neurons to be more easily visualized in the Bessel images than in the Gaussian image stacks (white arrowheads, *Figure 7F,H*). These data demonstrate that Bessel focus scanning improves imaging throughput of two-photon fluorescence microendoscopy by at least 10 × while improving lateral resolution and producing sharper images of synapses in brain tissue.

## In vivo volumetric microendoscopic imaging of synapses with Bessel focus scanning

We next tested the performance of our volumetric microendoscopic imaging method in the mouse brain in vivo. A polyimide cannula (*Bocarsly et al., 2015*) was implanted into brains of Thy1-GFP-M

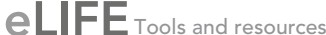

**Figure 7.** Bessel focus scanning improves imaging throughput and lateral resolution of two-photon fluorescence microendoscopy in fixed brain tissue. (**A**) Left: GABAergic neurons in a 370 × 370 × 55 µm³ volume color-coded by depth, imaged via 12 2D scans of a Gaussian focus through GRIN No. 4. Right: A single 2D scan of a Bessel focus imaged all the neurons. Post-GRIN lens power: 20 mW for Gaussian, 41 mW for Bessel. (**B**) GABAergic neurons in a 220 × 220 × 60 µm³ volume imaged through GRIN No. 5: 13 2D scans of a Gaussian focus (21 mW) and a single 2D scan of Bessel focus (55 mW). (**C**) GABAergic neurons in a 510 × 510 × 70 µm³ volume imaged through GRIN No. 6: 15 2D scans of a Gaussian focus (13 mW) and a single 2D scan of Bessel focus (50 mW). (**D**) Pyramidal neurons and their neurites across a 510 × 510 × 72 µm³ volume imaged through GRIN No. 6: 19 2D scans of a Gaussian focus (39 mW) and a single 2D scan of a Bessel focus (100 mW). (**E**) Pyramidal neuron neurites in a 370 × 370 × 56 µm³ volume imaged through GRIN No. 4: 15 2D scans of a Gaussian focus (45 mW) and a single 2D scan of Bessel focus (94 mW). (**G**): Pyramidal neuron dendrites in a 220 × 220 × 56 µm³ volume imaged through GRIN lens No. 5: 15 2D scans of a Gaussian focus (34 mW) and a single 2D scan of a Bessel focus (95

*Figure 7 continued on next page*

*Figure 7 continued*

mW). (**F, H**) Zoomed-in views of dendrites from the cyan boxes in (**E, G**). White arrowheads indicate dendritic spines that were more easily visualized by Bessel focus. Scale bar: 60 μm.

DOI: https://doi.org/10.7554/eLife.40805.008

mice, in order to hold the 1 mm doublet GRIN lens above the CA1 region of the hippocampus during imaging sessions (*Figure 8*, *Figure 9*). To image the brain at different depths, we moved the animal with the implanted GRIN lens axially, thus changing the image WD. The axial shift of the focus within the brain was then calculated from the relationship between the image and sample WDs as characterized independently with beads (*Figure 2C*).

In these mice, GFP was expressed in a sparse subset of CA1 pyramidal neurons, which allowed us to test for the ability of our microendoscopy system to image somata, dendrites, and axons, as well as their synaptic terminals in vivo. A volume of 370 μm × 370 μm × 53 μm obtained by scanning the Gaussian focus over 14 2D planes was imaged by a single 2D scanning of a Bessel focus (NA = 0.3, axial FWHM = 53 μm, *Figure 8A*). Similar to the results in the fixed brain slices, all somata and neurites within this volume were imaged by 2D scanning the axially elongated Bessel focus, which also provided superior lateral resolution over the Gaussian focus and allowed dendritic spines to be more readily detected in vivo (*cf.*, white arrowheads, *Figure 8B*). Moving the animal and GRIN lens down by 400 μm shifted the focus upward by 70 μm into a more dorsal layer of the hippocampus, where the structures were dominated by neuronal processes of the CA1 neurons. Bessel focus

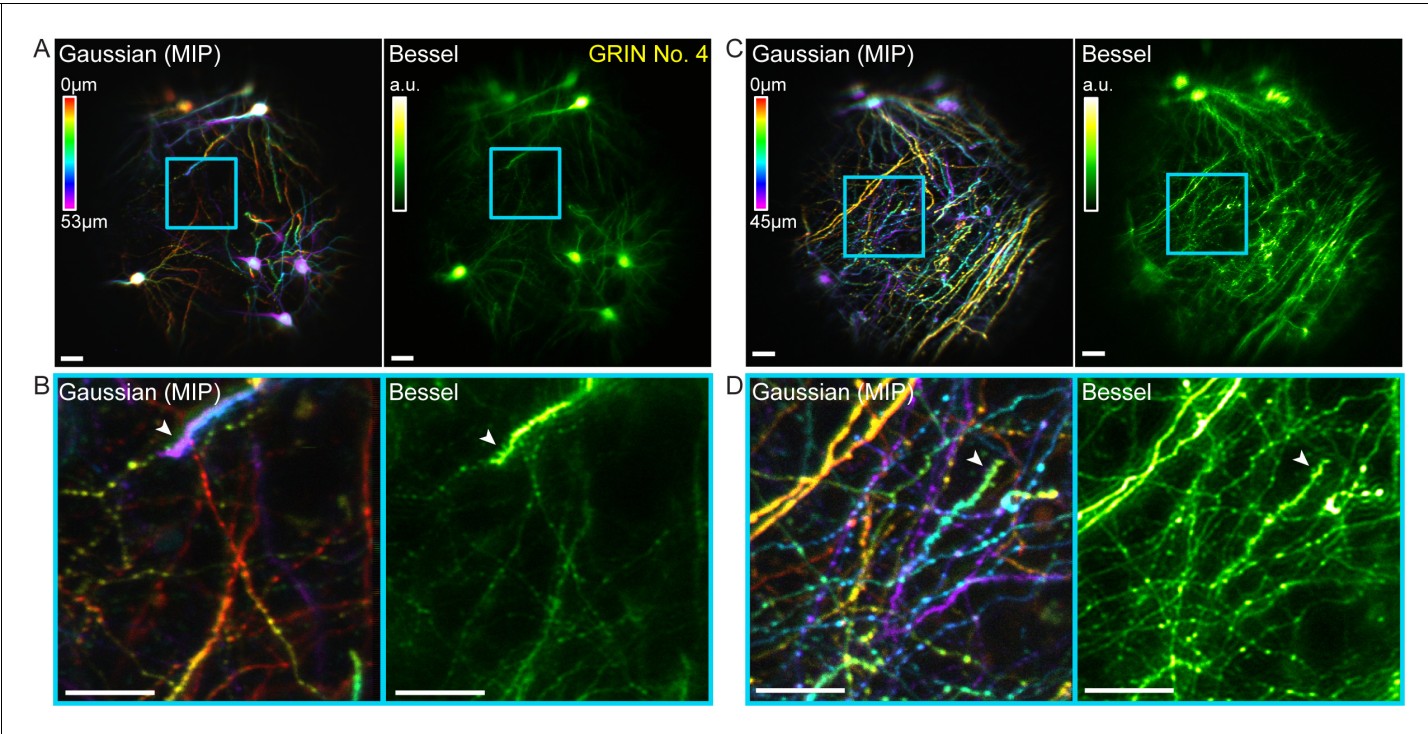

**Figure 8.** Bessel focus scanning allows volumetric microendoscopic imaging of dendritic spines and axonal boutons in mouse hippocampus in vivo. (**A**) Left panel: Hippocampal neurons and neurites within a 370 μm × 370 μm × 53 μm volume color-coded by depth, imaged via 14 2D scans of a Gaussian focus through a 1-mm-diameter doublet GRIN lens. Right panel: Image obtained by a single 2D scan of a Bessel focus (NA: 0.3, axial FHWM: 53 μm). Post-GRIN lens power: 46 mW for Gaussian, 84 mW for Bessel. Image WD: 0 μm. (**B**) Zoomed-in views of dendritic spines and axonal boutons within the cyan box in (**A**). (**C**) Left panel: Hippocampal axons and dendrites within a 370 μm × 370 μm × 45 μm volume color-coded by depth, imaged via 11 2D scans of a Gaussian focus. Right panel: Image obtained by a single 2D scan of the Bessel focus. Post-GRIN lens power: 60 mW for Gaussian, 110 mW for Bessel. Image WD: 400 μm. (**D**) Zoomed-in views of dendritic spines and axonal boutons within the cyan box in (**C**). White arrowheads point to dendritic spines. Scale bar: 20 μm.

DOI: https://doi.org/10.7554/eLife.40805.009

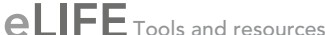

**Figure 9.** Immunohistochemistry on brain sections surrounding the GRIN lens implantation site. (**A**) and (**B**) Representative brain slices labeled with GFAP antibody and Nissl staining, respectively. (**C**) Consecutive sections of a brain alternately stained against GFAP (odd number labels) and Nissl staining (even number labels). The green bars indicate the size of the field of view for this GRIN lens (GRIN lens No. 4). Scale bar: 1 mm.

DOI: https://doi.org/10.7554/eLife.40805.010

scanning again provided sharper volumetric images of axonal boutons and dendritic spines (*Figure 8C,D*). Combined with the increase of imaging throughput, two-photon fluorescence microendoscopy coupled with the Bessel module is clearly a more advantageous way to image deeply buried albeit sparsely labeled brain structures at synaptic resolution.

## Volumetric imaging of population dynamics of lateral hypothalamic GABAergic neurons during different metabolic states in vivo

Two-dimensional scanning with a Bessel focus can also be used to monitor population activity dynamics of neurons distributed in 3D within deeply buried nuclei in vivo. In addition to increased image throughput, the axial extension of the Bessel focus also makes the microendoscopic imaging process resistant to axial-motion-induced artifacts (*Lu et al., 2017*; *Lu et al., 2018*). As an example, we applied Bessel focus scanning microendoscopy to study the population activity of GABAergic neurons in the lateral hypothalamus (LH) of awake mice during native metabolic conditions that mimicked hunger and satiety.

With the LH located ~5 mm below the dorsal surface of the brain and our goal of studying cellular-level population activity, we chose to use the 0.5 mm singlet GRIN lens, because it had less insertion damage while providing cellular resolution over a relatively large FOV. A Cre-dependent GCaMP6s adeno-associated virus was injected and GRIN lenses were implanted into the LH of Vgat-Cre mice to visualize calcium activity in GABAergic neurons of the LH. One month after surgery, the GCaMP6s-expressing neurons were imaged during two sessions daily with 10 hr between sessions to record neuronal activity. One set of imaging experiments was carried out at the end of the light cycle, which is a period of increased motivation to eat; the other set was performed at the end of the dark cycle, when mice are usually sated and have decreased motivation to eat. We waited for three days before repeating the experiments to avoid any imaging-induced disruption of feeding and to ensure that mice returned to normal food intake schedules. During each imaging session with head-fixed awake mice, we recorded serial 3D scans with a Gaussian focus (axial FWHM 19 μm) to obtain anatomical information. We subsequently used a Bessel focus (axial FWHM 63 μm) to record the calcium transients from all the GABAergic neurons ($N$ = 43 and 51 neurons in Mouse 1 and 2, respectively) across a 310 μm × 310 μm × 63 μm volume for 30 min. With the absence of axial motion artifacts (e.g., *Videos 1* and *2*), the imaging data only needed to be registered in 2D, first with the TurboReg plugin (*Thévenaz et al., 1998*) in Fiji (*Schindelin et al., 2012*; *Schneider et al., 2012*) to correct rigid lateral motions, then by a non-rigid lateral motion correction method for optimal registration results (*Pnevmatikakis and Giovannucci, 2017*). Regions of interest (ROIs) corresponding to the somata of GABAergic neurons were then segmented by hand, and their corresponding calcium transients were extracted from the time-lapse data and quantified as ΔF/F (fluorescence change divided by baseline fluorescence signal).

Although the neuronal recordings were taken during different experimental sessions that were hours and days apart with mice freely behaving in between, the same neurons could be identified and tracked across sessions both in Gaussian stacks and Bessel images for each mouse (*Figure 10A,B*). This enabled us to directly compare the activity of the same neurons across different sessions while the animal was under distinct metabolic states (*Figure 10C–K*). We found that these GABAergic neurons exhibited frequent, mostly synchronous activity during both ends of light and dark cycles, as indicated by representative traces of calcium transients from selected neurons (*Figure 10C,D*). When mice were imaged at the end of light cycles, we observed increased activity, which was corroborated by quantitative analysis using the calcium

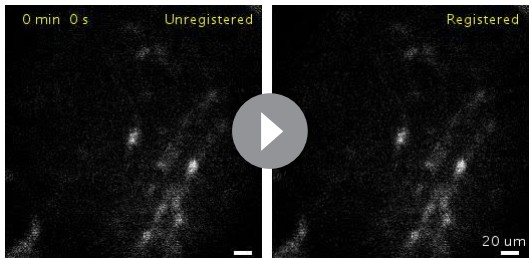

**Video 1.** Example volumetric microendoscopy recordings of GCaMP6s+ lateral hypothalamic GABAergic neurons in *Figure 10A* (Mouse 1). All the neurons within the 310 μm × 310 μm × 63 μm volume were in focus throughout the recording, indicating the absence of axial motion artifact via Bessel focus scanning. Left panel: unregistered stack; Right panel: image stack after correcting lateral motion with TurboReg plugin (*Thévenaz et al., 1998*) followed by a non-rigid lateral motion correction method (*Pnevmatikakis and Giovannucci, 2017*).
DOI: https://doi.org/10.7554/eLife.40805.011

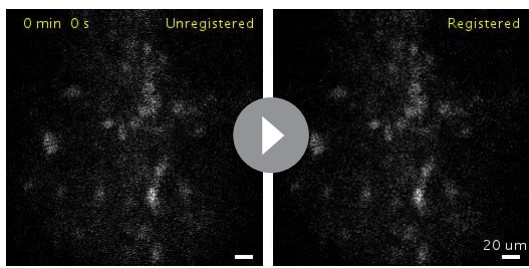

**Video 2.** Example volumetric microendoscopy recordings of GCaMP6s+ lateral hypothalamic GABAergic neurons in *Figure 10B* (Mouse 2). All the neurons within the 310 μm × 310 μm × 63 μm volume were in focus throughout the recording, indicating the absence of axial motion artifact via Bessel focus scanning. Left panel: unregistered stack; Right panel: image stack after correcting lateral motion with TurboReg plugin (*Thévenaz et al., 1998*) followed by a non-rigid lateral motion correction method (*Pnevmatikakis and Giovannucci, 2017*).
DOI: https://doi.org/10.7554/eLife.40805.012

transients from all detected neurons (*Figure 10E,G*). The activity of each neuron was quantified as the integrated value of its ΔF/F curve, while the activity synchrony between two neurons was calculated as the correlation coefficient (*R*) of their ΔF/F traces (*Figure 10F,H*). For both mice, the population activity and synchrony of these GABAergic neurons increased significantly from the end of dark cycles to the end of light cycles, that is from a satiety-like state to a hunger-like state (175 neurons, 3820 correlation coefficients, non-paired t-test, *Figure 10I–K*). As demonstrated in this example, 2D Bessel focus scanning microendoscopy enables high-throughput, simultaneous 3D imaging and subsequent analysis of population dynamics in deeply buried brain structures.

## Discussion

Volumetric imaging of neurons and neuronal activity at high spatial and temporal resolution is crucial to study information processing of neuronal circuits. Deeply buried brain nuclei can now be accessed optically via microendoscopy through micro-optical probes embedded in the brain. For structures such as hippocampus and dorsal lateral geniculate nucleus (*Mizrahi et al., 2004*; *Dombeck et al., 2010*; *Marshel et al., 2012*), a cannula with a glass bottom can be embedded to provide optical access after the removal of overlying tissue, and a microscope objective with a long working distance can be used for imaging. Such an approach, however, is only applicable to more superficial structures. For deeper structures, providing optical access for a conventional microscope objective requires too much tissue removal. For example, to image the lateral hypothalamus, which is 5 mm from the top surface of the mouse brain, passing the light cone of a 0.5-NA microscope objective requires 22 mm$^3$ of brain tissue to be removed, whereas an embedded 0.5-mm-diameter GRIN lens only displaces 1 mm$^3$ of brain tissue. Therefore, microendoscopy employing GRIN lenses is the preferred method for imaging deep nuclei of the brain. In both cases, care must be taken to ensure that inflammation has abated before imaging starts and that the physiology of the structures of interest is minimally affected by tissue removal or displacement.

Existing implementation of microendoscopy suffers from low resolution and/or low throughput. By scanning an axially elongated Bessel focus through selected commercially available GRIN lenses, we addressed both the issues of resolution and throughput. The intrinsically higher lateral resolution of Bessel foci allowed us to resolve dendritic spines and axonal boutons both ex vivo and in vivo, while improving imaging throughput up to 19×.

The axially extended two-photon fluorescence excitation in Bessel focus scanning microendoscopy also makes the imaging process immune to axial motion artifacts and allowed us to obtain high-quality functional imaging data from awake behaving mice without having to resort to complex post-processing. Because more energy is distributed in the side rings of a Bessel focus, the average power of the excitation light was increased by 1.5–4×, which remains within the safe range for two-photon fluorescence microscopy (*Podgorski and Ranganathan, 2016*). Although we only demonstrated the applications in relatively sparsely labeled samples, Bessel focus scanning microendoscopy can also be applied to more densely labeled samples, and computational methods (*Liu et al., 2012*; *Pnevmatikakis et al., 2016*; *Zhou et al., 2018*) can be used to demix activity from overlapping neurons. Finally, because Bessel focus scanning modules (*Lu et al., 2017*; *Lu et al., 2018*) can be added to existing microendoscopy setups without additional changes in data acquisition software, we expect this method to be easily adoptable for high-throughput synaptic resolution volumetric imaging of deep brain structures.

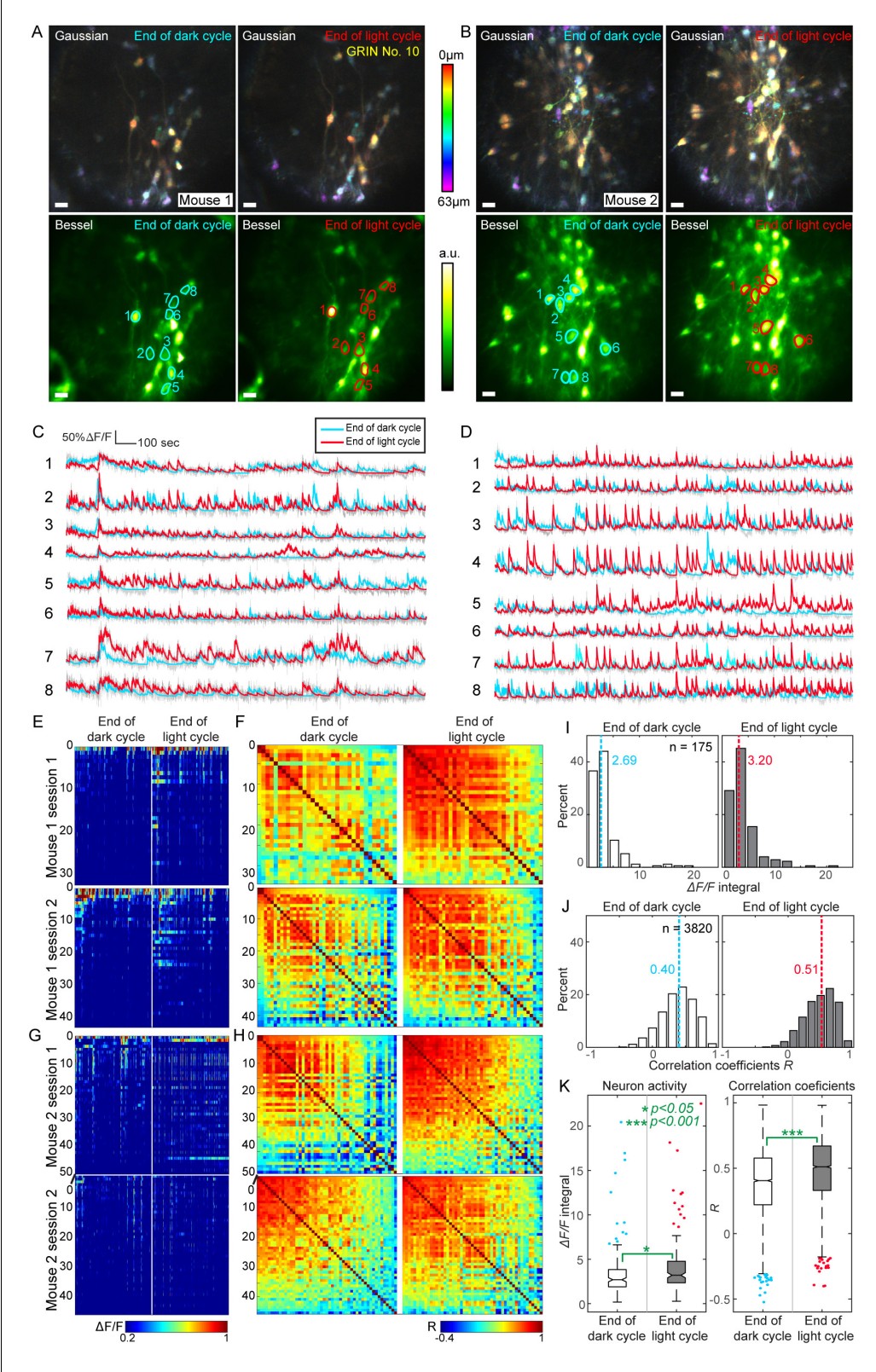

**Figure 10.** High throughput functional imaging via Bessel focus scanning enables monitoring of 3D population dynamics of GABAergic neurons in lateral hypothalamus during different metabolic states. (**A, B**) In two mice, Gaussian stacks were taken through a 0.5-mm-diameter singlet GRIN lens of neurons in a 310 µm × 310 µm × 63 µm volume (top panels, color-coded by depth), followed by functional volumetric imaging using Bessel focus scanning for 30 min to record spontaneous activity from these neurons at the ends of dark and light cycles (bottom panels show the time-averaged

*Figure 10 continued on next page*

*Figure 10 continued*

images). Post-GRIN lens power: 70 mW for Gaussian; 115 mW for Bessel; Axial FWHM: 19 µm for Gaussian; 63 µm for Bessel. Scale bar: 20 µm. (C,D) ΔF/F calcium transient traces of neurons outlined in A and B, respectively. (E) Raster plots of ΔF/F from all identified neurons in Mouse 1, sorted by their activity at the end of dark cycle. (F) Correlation coefficient (R) matrices between ΔF/F traces from all neuron pairs in E. (G–H) Raster plots and R matrices for Mouse 2. (I,J) Histograms of activity (integral of ΔF/F traces) and R. Dashed lines: medians. (K) Notched box whisker plots of neuronal activity and R, showing statistically significant differences in neuronal activity and their correlation at the ends of dark and light cycles (non-paired t-test). Outliers highlighted in the box-whisker plots were included in the statistical analysis.

DOI: https://doi.org/10.7554/eLife.40805.013

## Materials and methods

### Animals
All animal experiments were conducted according to the United States National Institutes of Health guidelines for animal research. Procedures and protocols were approved by the Institutional Animal Care and Use Committee at Janelia Research Campus, Howard Hughes Medical Institute. Male and female mice (C57BL/6J background, Strain 664, The Jackson Laboratory, ME, USA) aged two months and older were used in this study: Gad2-IRES-cre (*Network NNBCD, 2009*; *Taniguchi et al., 2011*), Thy1-GFP line M (*Feng et al., 2000*), Ai93(24107)×ACTB tTA×Kcnd2 IRES-Cre 3G5 (*Madisen et al., 2015*; *Li et al., 2010*), Vgat-cre (*Vong et al., 2011*). Prior to stereotaxic surgery, mice were group housed with littermates in temperature and humidity-controlled rooms with *ad libitum* access to water and rodent chow (PicoLab Rodent Diet 20, 5053 tablet, LabDiet/Land O'Lakes Inc., MO, USA) on a 12 hr reverse light/dark cycle (9 p.m.−9 a.m. light cycle and 9 a.m.− 9 p.m. dark cycle).

### Two-photon excitation fluorescence microendoscopy
A homebuilt two-photon excitation fluorescence microscope was used for microendoscopic imaging and was described in detail previously (*Ji et al., 2010*). 940 nm output from a femtosecond pulsed laser source with a built-in dispersion compensation unit (InSight DeepSee, Spectra Physics, CA, USA) was used to provide two-photon excitation. A 10×/0.45NA microscope objective (CFI Plan Apo Lambda 10X, Nikon, Tokyo, Japan) was used in conjunction with 0.5-NA singlet GRIN lenses, whereas a 4×/0.2NA objective (CFI Plan Apo Lambda 4X, Nikon) was used with doublet GRIN lenses to match their NA on the image side. Microscope control and data acquisition were executed using custom-written LabVIEW software.

### Alignment of GRIN lenses
Obtaining optimal image quality from GRIN-lens-based microendoscopes requires the GRIN lens to share the same optical axis as the microscope objective and the excitation light. For the in vitro characterization, the GRIN lens was held with a custom-built clip mounted onto a tilt/tip adjustment platform mount (KM100B, Thorlabs, NJ, USA). The tilt/tip platform was further mounted onto a motorized 3D translational stage (3DMS, Sutter Instrument Company, CA, USA), which allowed us to adjust the tilt/tip and translational position of the GRIN lens. Looking through the eyepiece and the 10×/0.45NA objective, we adjusted the tilt/tip of the GRIN lens so that the entire top surface of the GRIN lens came into focus simultaneously, indicating that the optical axes of the GRIN lens and the objective were parallel. Failure to do this causes the excitation light to experience severe optical aberrations and degradation of image quality. To make the GRIN lens and the objective coaxial, we switched to Bessel mode with the excitation light forming an annular illumination pattern when entering the GRIN lens. We then adjusted the translational position of the GRIN lens until the annular illumination transmitted through the lens remained symmetrical when the GRIN lens was translated axially, indicating that the GRIN lens was coaxial with the microscope objective. We then recorded the position of the GRIN lens within the eyepiece field of view (FOV). For in vivo experiments, the mouse was mounted on a tilt/tip platform and was adjusted below the 10×/0.45NA objective until the top surface of the GRIN lens was in focus. We then translated the mouse until the GRIN lens occupied the same position as recorded during the in vitro alignment. For doublet GRIN lenses, tip/tilt alignment was established using the 10×/0.45NA objective due to its smaller depth of

field; this objective was then replaced by the 4×/0.2NA objective prior to adjustment of translational position and the imaging experiments to follow.

## In vitro characterization of GRIN lenses

The imaging resolution and FOV of 10 commercially available GRIN lenses (GRINTECH GMbH, Inscopix, Go!Foton) were characterized. Prior to the experiment, precise alignment of each GRIN lens was completed as described above. For doublet GRIN lenses with 0.19 NA on the image side and 0.5 NA on the sample side, data were collected with the image WD within a 2000 μm range. For singlet GRIN lenses, data were collected with the image WD within a 400 μm range. For all the GRIN lenses tested, the sample WD versus image WD followed a linear relationship with a fitted slope $k$. For in vivo experiments, the change of the sample WD (i.e., focal shift in the brain), $\Delta d_{sample}$, was calculated as $\Delta d_{sample} = k \times \Delta d_{image}$, where $\Delta d_{image}$ is the change of image WD (e.g., the axial shift of the brain and the GRIN lens) and $k$ is the slope determined by fitting the image WD and sample WD relationship (*Figure 2C*).

## Stereotaxic surgery and brain slice preparation

GABAergic neurons were labeled by in vivo injection of AAV2/1-Syn-Flex-GCaMP6s (1:2 diluted by PBS from stock solution with the titer of $2.7 \times 10^{13}$ GC/ml made by viral services at Janelia Research Campus, Howard Hughes Medical Institute) into the primary visual cortex of the Gad2-ires-Cre mouse (JAX, stock#010802). Thirty nanoliters of viral solution was injected at the following coordinate: Bregma, −3.80 mm; midline:+2.50 mm; depth 0.5 mm. Four weeks later, mice were perfused by 4% paraformaldehyde and the brain was cut using a vibratome (Leica VT1200S) into 100 μm-thick brain slices. Brain slices with positively labeled neurons were mounted with mounting medium (Vectashield, Vector Laboratory) onto slides for long-term preservation of the fluorescent signal. The same slice preparation procedure was followed for Thy1-GFP-M mice (JAX stock #007788).

## Stereotaxic surgery for in vivo imaging

All surgeries were performed using a stereotaxic apparatus (Model 1900, David Kopf Instruments, CA, USA) and aseptic technique. A stereotaxic cannula holder (SCH_2.5, Doric Lenses Inc., Québec, Canada) attached with a custom adapter (OD = 15.8 mm; ID = 7.9 mm, Doric Lenses Inc.) to a Model 1900–54-A tool block assembly (David Kopf Instruments) was used to insert a polyimide guide cannula into the brain.

Guide cannulae (*Bocarsly et al., 2015*) (Doric Lenses Inc) were custom designed for hippocampus surgeries. The cannula body was made of a thin-walled polyimide tube (0.25 μm wall thickness). The sample-side end was sealed with a 0.13-mm-thick cover glass. The open end of the tube was surrounded by a cap (2.5-mm-diameter body, 3.5-mm-diameter base) that allowed the cannula to be held during implantation and provided a large surface for secure attachment to the skull by dental cement. To fit the 1-mm-diameter GRIN lenses, the polyimide tubes had an inner diameter of 1.07 mm and an outer diameter of 1.12 mm. The length of the polyimide tube was selected such that, with the bottom of the cap contacting the skull, the glass bottom of the cannula was positioned above the deep nuclei of interest (lengths from the bottom of the cap to the glass bottom: 1.55 mm).

### Hippocampus surgery

Thy1-GFP line M mice were anesthetized with 1.5% isoflurane and the skull was exposed. After both medial-lateral and anterior-posterior alignment of mouse head at Bregma, the coordinate of implantation center on the right hemisphere (Bregma, −2.06 mm; midline:+2.71 mm) was marked. The animal was then tilted mid-laterally by 10 degrees toward the left hemisphere to slightly compensate for the oblique surface of the hippocampus. Then a 1.5-mm-diameter craniotomy was made. The dura was removed with forceps, and aspiration with a 27G blunt needle was used to slowly remove the cortex and clear a 1.2-mm-diameter pathway within the craniotomy. Each aspiration (50–100 μm-thick tissue removed per aspiration) was followed by repeated irrigation with saline until bleeding stopped completely. These steps continued until the external capsule was exposed. The cortex and topmost layers of the external capsule were carefully aspirated with a 30G blunt needle. After bleeding stopped, a guide cannula was inserted and cemented to the skull using C&B Metabond Cement

System (Parkell, Inc., NY, USA). A titanium head-post was attached to the skull with cyanoacrylate glue and Metabond. The open end of the cannula was sealed with a small piece of Parafilm covered by Kwik-Sil (World Precision Instruments, LLC, FL, USA), and after the silicone cured, mice recovered from anesthesia. After one month of recovery from the surgery and habituation to head fixation, we inserted a 1-mm-diameter doublet GRIN lens into the cannula for in vivo imaging in anesthetized mice and removed the GRIN lens at the end of imaging. The cannula was resealed with Parafilm and Kwik-Sil to prevent debris from entering the guide tube.

## Lateral hypothalamus surgery

Surgeries were performed as previously described (*Bocarsly et al., 2015*), except 0.5-mm-diameter singlet GRIN lenses (Go!Foton Corporation) were directly implanted into the brains of Vgat-Cre mice (JAX stock #028862) without guide cannulae (Bregma, −1.4 mm; midline,+0.85 mm; dorsal surface: −5.0 mm; 50 nl AAV2/1-Syn-Flex-GCaMP6s virus) (*Bocarsly et al., 2015*; *Chen et al., 2013*). Mice were returned to their reverse light cycle holding room after recovery from anesthesia. After post-surgical recovery and habituation to head fixation for at least four weeks, neuronal activity was monitored in awake mice at the end of the light cycle (9 a.m.) (*i.e.*, after a period of low food intake) and at the end of the dark cycle (11 p.m.) (*i.e.*, after a period of high food intake). Each mouse was imaged during two sessions daily (light cycle end and dark cycle end) with 10 hr between sessions to record neuronal activity when the mice were in native metabolic states similar to hunger and satiety (light cycle end and dark cycle end, respectively). These two sessions were repeated twice with at least 3 days between repetitions.

## Immunohistochemistry characterization of the implant site

After data acquisition, mice were anesthetized with an overdose of isoflurane and transcardially perfused with phosphate buffered saline (PBS) followed by 4% paraformaldehyde in PBS. Brains were postfixed in the same fixative overnight and stored at 4℃ for later histological process. Brains were coronally sectioned (50 μm thickness) with a Leica VT1200S vibratome. Alternate brain sections were either immunostained with the astrocyte marker, anti-glial fibrillary acidic protein (GFAP, Sigma Aldrich), or stained with NeuroTrace 530/615 red fluorescent Nissl Stain (ThermoFisher). Consecutive brain sections encompassing the implant site were mounted on glass slides using VECTASHIELD mounting medium with DAPI (Vector Laboratories) and coverslipped for imaging with TissueFAXS 200 (TissueGnostics) at wide-field mode using a 20× objective.

In the example brain (*Figure 9*), only the thin layer of tissue right next to the implant site exhibited enhanced staining for GFAP, in agreement with a previous study (*Bocarsly et al., 2015*). Nissl staining showed some tissue deformation in the hippocampal structures below the implant site. However, CA1 pyramidal neurons within the imaging FOV (indicated by the green bars in *Figure 9*) maintained laminar distribution. This is consistent with the in vivo imaging results in *Figure 8* (same brain as in *Figure 9*), in which the GFP[+] dendrites and axons were found to have normal morphology and were devoid of signs of cell death (e.g., fragmented axons).

## Image processing and analysis

Imaging data were processed with Fiji (*Schindelin et al., 2012*; *Schneider et al., 2012*) and MATLAB (Mathworks, MA, USA). Raw images were presented with standard color bars in Fiji without further processing, with the exception of Bessel images in *Figure 8A,B* in which, due to the overwhelmingly bright somata, gamma correction ($\gamma = 0.7$ and $\gamma = 0.85$, respectively) was applied to enhance the visibility of dendrites. Functional images were registered with TurboReg plugin (*Thévenaz et al., 1998*) in Fiji for rigid motion and with a non-rigid motion correction algorithm in MATLAB (*Pnevmatikakis and Giovannucci, 2017*). ROIs were selected and outlined manually in Fiji. The averaged fluorescent signal within the ROI was then extracted with custom MATLAB codes (available online as a source code file) and used to calculate calcium transients. The traces in *Figure 10* were smoothed with a constrained deconvolution (CD) method in MATLAB (*Pnevmatikakis et al., 2016*).

## Acknowledgements

The authors thank Daniel Dombeck and Christine Grienberger for mouse preparation demos, and Amy Hu for help with immunohistochemistry. GM, YL, WJ, RL, JD and NJ were supported by Howard Hughes Medical Institute. GM and NJ were supported by National Institutes of Health (NINDS U01NS103489). SS and YA were supported by the National Institute on Drug Abuse Intramural Research Program, U.S. National Institutes of Health (NIDA/IRP/NIH).

## Additional information

### Competing interests

Rongwen Lu, Na Ji: The Bessel focus scanning intellectual property has been licensed to Thorlabs, Inc by HHMI. The other authors declare that no competing interests exist.

### Funding

| Funder | Grant reference number | Author |
| --- | --- | --- |
| Howard Hughes Medical Institute | | Guanghan Meng<br>Yajie Liang<br>Wan-chen Jiang<br>Rongwen Lu<br>Joshua Tate Dudman<br>Na Ji |
| National Institute of Neurological Disorders and Stroke | U01NS103489 | Guanghan Meng<br>Na Ji |
| National Institute on Drug Abuse | | Sarah Sarsfield<br>Yeka Aponte |

The funders had no role in study design, data collection and interpretation, or the decision to submit the work for publication.

### Author contributions

Guanghan Meng, Data curation, Software, Formal analysis, Validation, Investigation, Visualization, Methodology, Writing—original draft, Writing—review and editing; Yajie Liang, Validation, Investigation, Methodology, Writing—original draft, Writing—review and editing; Sarah Sarsfield, Resources, Methodology, Writing—original draft, Writing—review and editing; Wan-chen Jiang, Investigation, Methodology, Writing—review and editing; Rongwen Lu, Software, Methodology, Writing—review and editing; Joshua Tate Dudman, Yeka Aponte, Resources, Supervision, Funding acquisition, Methodology, Writing—review and editing; Na Ji, Conceptualization, Resources, Supervision, Funding acquisition, Methodology, Writing—original draft, Project administration, Writing—review and editing

### Author ORCIDs

Joshua Tate Dudman [ID] http://orcid.org/0000-0002-4436-1057
Yeka Aponte [ID] http://orcid.org/0000-0002-5967-2579
Na Ji [ID] https://orcid.org/0000-0002-5527-1663

### Ethics

Animal experimentation: All animal experiments were conducted according to the United States National Institutes of Health guidelines for animal research. Procedures and protocols were approved by the Institutional Animal Care and Use Committee at Janelia Research Campus, Howard Hughes Medical Institute (protocol number: 16-147)

### Decision letter and Author response

Decision letter https://doi.org/10.7554/eLife.40805.019
Author response https://doi.org/10.7554/eLife.40805.020

# Additional files

## Supplementary files

• Transparent reporting form
DOI: https://doi.org/10.7554/eLife.40805.014

## Data availability

Almost all data needed to evaluate the conclusions in the paper are present in the paper or the supplementary materials; Raw image data for Figs. 2-10 are available from Dryad, 10.5061/dryad.pr4t978

The following dataset was generated:

| Author(s) | Year | Dataset title | Dataset URL | Database and Identifier |
|---|---|---|---|---|
| Meng G, Liang Y | 2019 | Data from: High-throughput synapse-resolving two-photon fluorescence microendoscopy for deep-brain volumetric imaging in vivo | http://dx.doi.org/10.5061/dryad.pr4t978 | Dryad, 10.5061/dryad.pr4t978 |

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
