## [Decision Letter]

Thank you for submitting your article "High-throughput synapse-resolving two-photon fluorescence microendoscopy for deep-brain volumetric imaging in vivo" for consideration by *eLife*. Your article has been reviewed by two peer reviewers, including David Kleinfeld as the Reviewing Editor and Reviewer #2, and the evaluation has been overseen by Eve Marder as the Senior Editor. The other reviewer opted to remain anonymous.

All in all this is a fine and important technical addition to field of neuroimaging. There is overlap among the reviews. We ask that you supply histological data, as suggested by the Reviewing Editor; this may entail new measurements. We further ask that you revise the text to comment on the utility of a cannula versus a GRIN lens for imaging neurons in CA1, as suggested by both reviewers.

*Reviewer #1:*

The paper demonstrates the feasibility and advantages of 2P Bessel volumetric imaging through GRIN lenses. There is a nice comparison of the various available GRIN lenses with thorough characterization of field of view size and axial resolution. Then there are examples of recorded neurons in hippocampus, hypothalamus and striatum showing that the technique is feasible in-vivo. For sparsely expressed indicators this would be extremely effective and efficient.

A GRIN lens is not needed for imaging dorsal CA1 hippocampal neuronal cell bodies or their dendrites or dorsal striatum. Imaging can be performed with a coverslip at the bottom of a cannula without any intervening lens. I would be curious to compare the performance of the Bessel beam technique through a GRIN lens and without a GRIN lens in these preparations.

*Reviewer #2:*

This is a very useful paper from Na Ji's group that expands on their 2017 Nature neuroscience paper on two photon imaging with axially extended beam. The key features are: (1) An assessment of different commercial GRIN lenses for in vivo neuroimaging; (2) Extension to GRIN lens scanning; and (3) Extension of Bessel beams to another imaging mode. While Bessel beams are primarily useful for the case of sparsely labeled cells, this is a common and reasonable limit. I have one substantive suggestion a number of editorial suggestions.

Major:

This paper will gain in value if the authors included histology of the recording region. Specifically, I would add alternate brain sections that span the recording site. One with just an image of the GCaMP along a measure of damage – such as GFAP stain (maybe with a red label) and alternating with Nissl stained sections -. This will allows users to gauge damage, a topic that the authors already allude to by their statement; "… we chose to use the 0.5-mm singlet GRIN lens, because it had less insertion damage while providing cellular resolution over a relatively large FOV." This critical issue needs to be justified. It could be that the density of labeled neurons is reduced in the measurements volume under the lens. Or not. We are interested in typical – not "typical best", so that readers understand the limitations of measurements with GRIN lenses.

---

## [Author Response]

All in all this is a fine and important technical addition to field of neuroimaging. There is overlap among the reviews. We ask that you supply histological data, as suggested by the Reviewing Editor; this may entail new measurements. We further ask that you revise the text to comment on the utility of a cannula versus a GRIN lens for imaging neurons in CA1, as suggested by both reviewers.

We have included additional histology data. We also added comments on cannula versus GRIN lens, as well as tissue damage, to the Discussion section:

“For structures such as hippocampus and dorsal lateral geniculate nucleus, after the removal of overlaying tissue (Mizrahi et al., 2004; Dombeck et al., 2010 and Marshel et al., 2012), a cannula with a glass bottom could be embedded to provide optical access, and a microscope objective with long working distance used for imaging. […]In both cases, because of damages caused by tissue removal, care must be taken to ensure that the inflammation has abated before imaging starts and that the physiology of the interested structures is minimally affected by tissue removal.”

Reviewer #1:

The paper demonstrates the feasibility and advantages of 2P Bessel volumetric imaging through GRIN lenses. There is a nice comparison of the various available GRIN lenses with thorough characterization of field of view size and axial resolution. Then there are examples of recorded neurons in hippocampus, hypothalamus and striatum showing that the technique is feasible in-vivo. For sparsely expressed indicators this would be extremely effective and efficient.A GRIN lens is not needed for imaging dorsal CA1 hippocampal neuronal cell bodies or their dendrites or dorsal striatum. Imaging can be performed with a coverslip at the bottom of a cannula without any intervening lens. I would be curious to compare the performance of the Bessel beam technique through a GRIN lens and without a GRIN lens in these preparations.

We thank the reviewer for the suggestion. Please see the comments above comparing cannula with GRIN lens embedding.

Reviewer #2:

*This is a very useful paper from Na Ji's group that expands on their 2017 Nature neuroscience paper on two photon imaging with axially extended beam. The key features are: (1) An assessment of different commercial GRIN lenses for* in vivo *neuroimaging; (2) Extension to GRIN lens scanning; and (3) Extension of Bessel beams to another imaging mode. While Bessel beams are primarily useful for the case of sparsely labeled cells, this is a common and reasonable limit. I have one substantive suggestion a number of editorial suggestions.*

We thank the reviewer for his thorough review and have modified the manuscript following his instructions.

Major:This paper will gain in value if the authors included histology of the recording region. Specifically, I would add alternate brain sections that span the recording site. One with just an image of the GCaMP along a measure of damage – such as GFAP stain (maybe with a red label) and alternating with Nissl stained sections -. This will allows users to gauge damage, a topic that the authors already allude to by their statement; "… we chose to use the 0.5-mm singlet GRIN lens, because it had less insertion damage while providing cellular resolution over a relatively large FOV." This critical issue needs to be justified. It could be that the density of labeled neurons is reduced in the measurements volume under the lens. Or not. We are interested in typical – not "typical best", so that readers understand the limitations of measurements with GRIN lenses.

We agree with the reviewer completely that we need to make sure that accompanying inflammation of the brain tissue near the implantation site have abated at the time of imaging experiments and the physiology of neurons under investigation is minimally impacted.

In an earlier publication (Bocarsly et al., 2015), we studied the time course of the inflammation response by immunohistochemically staining the glia markers GFAP and IBA1, following implantation of 0.5-mm singlet GRIN lens above lateral hypothalamus. As indicated by Figure 3 from Bocarsly et al., 2015, whereas inflammation surrounding the surgical site two weeks post implant was heightened, at four weeks post implant, the inflammation was largely gone. The neurons below the implantation site were found to have normal response pattern, suggesting that they maintained normal physiology.

Based on this prior work, all the in vivo imaging experiments in this manuscript were carried out 30-70 days post implantation of guide tube or GRIN lens. We have added in the Materials and methods section the time between surgery and imaging to highlight this fact.

Following the reviewer’s comment, we further carried out immunohistochemistry experiments on two brains with 1-mm-diameter GRIN lens implantation above hippocampus (Figure 9 and Author response image 1. These brains were fixed after imaging in 2017 and sectioned for immunohistochemistry in October 2018. (We did not carry out these experiments on the brains used for lateral hypothalamus imaging. Because these brains were already sectioned and coverslipped back in 2017, we are not confident that these sections maintain their antigenicity toward GFAP antibody. We expect that they would give rise to results similar to those in Figure 3 from Bocarsly et al., 2015, as the same brain area, lateral hypothalamus, was imaged.)

**Author response image 1. respfig1:** Immunohistochemistry on brain sections surround the GRIN lens implantation site. (**A**) and (**B)** Representative brain slices labeled with GFAP antibody and Nissl staining, respectively. (**C**) Consecutive sections of a brain alternately stained against GFAP (even number labels) and Nissl staining (odd number labels). Scale bar: 1 mm.

The results in Figure 9 are in agreement with our previously published results, with only the thin layer of tissue right next to the implant site exhibiting enhanced staining for glial fibrillary acid protein (GFAP). The Nissl staining showed tissue deformation in the hippocampal structures below the implant site. However, CA1 pyramidal neurons within the imaging FOV (indicated by the green bars in Figure 9) maintained laminar distribution. This is consistent with the in vivo imaging results in Figure 8 (same brain as in Figure 9), where the GFP+ dendrites and axons were found to have normal morphology and were devoid of signs of cell death (e.g., fragmented axons).

The immunohistochemistry results from the 2nd brain (Author response image 1) are similar, despite some tearing during sectioning. The hippocampal structures below the implant site showed less deformation than the brain in Figure 3 from Bocarsly et al., 2015 (thus, the results are typical, rather than “typical best”).

In the revised manuscript, we added a section in Materials and methods “Immunohistochemistry characterization of the implant site” and included discussions on tissue damage.